# Comparison of open anterior component and open transversus abdominus release in repair of large subcostal hernias

**Antonio Espinosa-de-los-Monteros**[1]*, **Daniela Fernandez-Alva**[1], **Rodrigo Alejandro Solis-Reyna**[1], **Cesar Alberto de-la-Garza-Elizondo**[1], **Joseph Vazquez-Guadalupe**[1], **Oscar Emmanuel Posadas-Trujillo**[2], **Flavio Enrique Diaz-Trueba**[2]

1 Department of Plastic Surgery, Instituto Nacional de Ciencias Medicas y Nutricion Salvador Zubiran, Mexico City, Mexico, 2 Department of General Surgery, Instituto Nacional de Ciencias Medicas y Nutricion Salvador Zubiran, Mexico City, Mexico

* aedlms@hotmail.com

## Abstract

### Background

Large subcostal incisional hernias are considered as complex defects, and a few different approaches have been described for repair. The purpose of this comparative cross-sectional study is to evaluate the outcomes of patients with large subcostal incisional hernias treated with either the open anterior components separation technique (ACS) or with the open transversus abdominis release technique (TAR).

### Methods

From the database of patients with large complex incisional hernias who underwent abdominal wall reconstruction with open techniques between April 2007 and October 2022 at our institution, on May 25th, 2023 we identified those whose hernias were located in the subcostal areas and who underwent reconstruction with a components separation technique and mesh. Perioperative variables and outcomes were compared between the patients with large subcostal hernias who underwent abdominal wall reconstruction with either the ACS or the TAR techniques.

### Results

Thirty-one patients with large subcostal hernias were included in the study. ACS and intra-abdominal mesh was used in 11 patients; TAR and retro-muscular mesh was performed in 20 patients. More postoperative local abdominal wall complications were seen in patients who had ACS as opposed to TAR (55% vs 15%, p = 0.02). Hernia recurrence was more common in patients who had ACS as opposed to TAR (55% vs 5%, p = 0.008).

### Conclusions

More post-operative complications and recurrences were seen in patients who had ACS as opposed to TAR.

**Data Availability Statement:** All relevant data are within the paper and its Supporting Information files.

**Funding:** The authors received no specific funding for this work.

**Competing interests:** The authors have declared that no competing interests exist.

## Introduction

Patients with history of hepato-pancreato-biliary (HPB), splenic or foregut surgical procedures performed through open subcostal incisions may develop incisional hernias [1]. These defects are considered as complex because 1) their proximity to bone structures limits the mobility of the abdominal wall muscles and interferes with adequate mesh overlap and fixation, 2) the separation of the muscles tends to occur in a cephalo-caudal direction, which is opposed to the typical latero-medial direction that the abdominal wall muscles can be moved using different surgical techniques, and 3) muscle atrophy is commonly associated due to denervation occurring from previous surgeries [2]. Small to medium-size subcostal incisional hernias can be successfully approached with minimally invasive techniques, but larger defects are more commonly treated with open techniques [3–5]. In the setting of large midline incisional hernias, two surgical procedures that have shown success in achieving full muscle closure are the anterior components separation technique (ACS) and the transversus abdominis release (TAR). Commonly, these procedures are coupled with mesh placement in order to reduce recurrence rates. In the setting of large incisional hernias located in the subcostal areas, little information exists with regards to these techniques and their comparison in terms of post-operative outcomes. The purpose of this study is to evaluate the outcomes of patients with large subcostal incisional hernias treated with open techniques, either ACS or TAR, and identify factors associated with the development of postoperative complications and recurrence.

## Materials and methods

This retrospective comparative cross-sectional study was approved by the Committee of Ethics in Research of the National Institute of Medical Sciences and Nutrition with the reference number 4447. From the database of patients with large complex incisional hernias who underwent abdominal wall reconstruction with open techniques between April 2007 and October 2022 at our institution, on May 25th, 2023, we identified those whose hernias were located in the subcostal areas and who underwent reconstruction with a components separation technique and mesh. For every patient, we recorded the following data from their medical chart: age, body mass index (BMI), history of smoking, diabetes mellitus, use of immunosuppressants, chronic obstructive pulmonary disease (COPD), history of previous infection and/or healing by secondary intention in the herniated site, history of previous hernia repairs in the same subcostal location, vertical and horizontal hernia size, rectangular area, elliptical area, intra-abdominal volume, herniated volume, volume ratio calculated as the herniated volume divided by the intra-abdominal volume, loss of domain ratio (LOD ratio) calculated as the herniated volume divided by the sum of herniated plus intra-abdominal volumes, enterotomies or any other source of contamination during the surgery, need for mesh removal, technique used for hernia repair (i.e. ACS or TAR), type of mesh used, in-hospital stay, morbidity rate including abdominal wall, intra-abdominal and systemic complications, re-operations needed to treat postoperative complications, total follow-up, and recurrence. During data collection, the authors had access to information that could identified individual participants, such as their full name. Written informed consent had been obtained from all participants to undergo their surgical procedure and for possible future research. The description of both surgical techniques for subcostal hernias has been published before by us [6,7], with the ACS technique being performed between April 2007 and February 2015, and the TAR technique being performed between February 2015 and October 2022. Briefly, in patients undergoing the ACS technique, the external oblique muscle on the same side as the hernia is separated from the internal oblique muscle laterally down to the level of the visible blood vessels to the external oblique muscle approximately at the level of the posterior axillary line. Also, the contralateral

external oblique muscle is separated both from the rectus abdominis and from the internal oblique muscles. A 30x30 cm coated mesh (Physiomesh, Ethicon or Parietene composite, Medtronic) was placed intra-peritoneally with an overlap of at least 5 cm to each side of the muscle borders and fixated with interrupted 0 polydioxanone (PDS, Ethicon) sutures with a separation of 2 cm between them. The hernia defect was subsequently closed with 0 polypropylene (Prolene, Ethicon) sutures approximating the lateral border of the rectus abdominis muscle to the medial border of the internal oblique and transversus abdominis muscles. In patients undergoing the TAR technique, we performed one of two possible approaches. The midline approach involved performing an incision along the medial border of the rectus abdominis fascia on the same side as the hernia, separating the posterior rectus sheath from the rectus muscle, incising the posterior lamella of the internal oblique muscle to expose the medial fibers of the transversus abdominis muscle both cephalically and caudally to the hernia defect, and separating the transversus abdominis muscle from the transversalis fascia laterally to the level of the quadratus lumborum muscle in the upper two thirds of the abdomen and to the psoas muscle in the lower third of the abdomen. In these patients, the contralateral rectus abdominis muscle was also separated from its underlying posterior rectus sheath, and the limit of the dissection was the emergence of the anterior cutaneous nerves. The other approach, a direct subcostal approach as described by San Miguel-Mendez et al, involved making the skin incision in the previous scar at the subcostal area [8]. In these patients, the posterior surface of the transversus abdominis muscle in the lateral border of the hernia defect was separated from the transversalis fascia along the entire length of the abdominal wall, and laterally to the level of the quadratus lumborum muscle in the upper two thirds of the abdomen and to the psoas muscle in the lower third of the abdomen. Also, in the medial border of the hernia, the posterior rectus sheath was separated from the posterior surface of the rectus abdominis muscle from lateral to medial. A cross-over maneuver to reach the contralateral rectus abdominis muscle was also performed on top of the extra-peritoneal extension of the falciform ligament, and the contralateral rectus abdominis fascia was incised medially to separate that rectus muscle from its posterior rectus sheath, and the limit of this dissection was the emergence of the anterior cutaneous nerves. In both approaches, the posterior rectus sheath in the same side as the hernia defect was approximated to the ipsilateral transversalis fascia with running 2–0 polyglactin (Vycril, Ethicon) sutures. Next, a non-coated mesh (Prolene, Ethicon; Soft mesh, BD; or Parietene, Medtronic) was placed with an overlap of at least 5 cm to each side of the muscle borders and fixated to the Cooper's ligament at the same side as the hernia, to the xyphoid and lateral to the lateral edge of the hernia defect using 2–0 polydioxanone (PDS, Ethicon) sutures. The lateral border of the rectus abdominis muscle was approximated to the medial border of the external oblique and internal oblique muscles using 0 polypropylene (Prolene, Ethicon) sutures (Fig 1).

Continuous variables are expressed as mean, except for in-hospital stay and total follow-up which are expressed as median and are compared using the Mann-Whitney's U test. Categorical variables are expressed as percentage and are analyzed using the Fisher's exact test. Significance was set at a p value lower than 0.05.

## Results

Between April 2007 and October 2022, a total of 245 open abdominal wall reconstructions for large complex incisional hernias were performed at our institution by the senior author. From them, 31 were performed for large defects located in subcostal areas and correspond to the universe of this study. Female patients accounted for 61%, while 39% were males. Mean age was 55 years-old (range 25 to 81 years-old), mean BMI was 28 kg/m$^2$ (range 22 to 36 kg/m$^2$) with

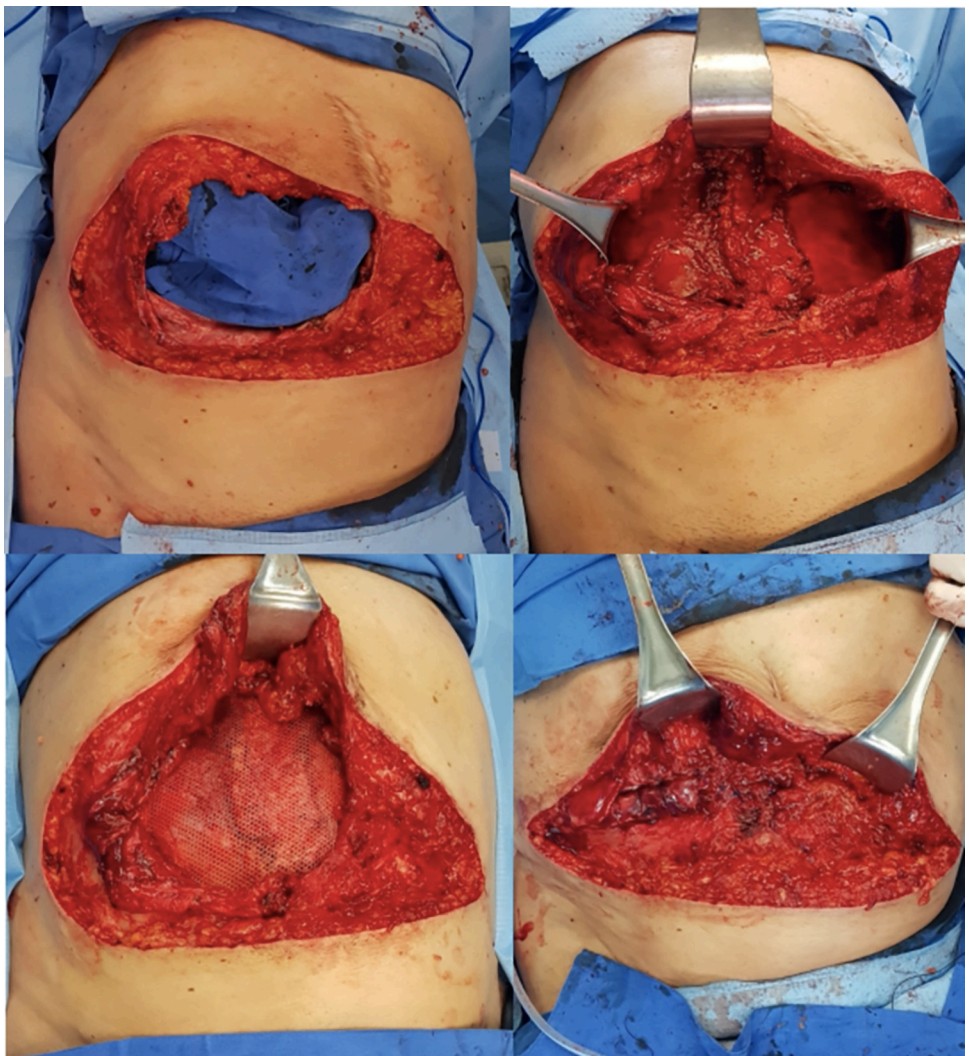

**Fig 1.** (Upper left): Right subcostal hernia measuring 26 x 23 cm in a patient with liver cirrhosis Child-Pugh A who had a right hepatectomy and a right colectomy one year before due to invasive hepatocellular carcinoma. (Upper right): A retromuscular plane behind the right transversus abdominis, the right rectus and the left rectus muscles has been developed, and closure between right posterior rectus sheath and right transversalis fascia has been achieved. (Lower left): A 30 x 30 cm large pore polypropylene mesh has been placed in the retromuscular plane. (Lower right): With the operating table flexed, horizontal closure of the right subcostal defect has been performed to provide restoration of the myofascial continuity of the abdominal wall.

26% of patients being obese (BMI 30 kg/m$^2$ or more). Smoking was present in 35%, diabetes mellitus in 13%, COPD in 3%, and immunosuppressants were used in 3%. In 23% of patients, the previous surgery had healed by secondary intention due to wound infection. History of previous hernia repair at the same subcostal area was present in 29% of patients. Mean hernia size was 12 x 12 cm (range 5 x 6 cm to 26 x 23 cm), with mean rectangular area of 164 cm$^2$ (range 30 to 592 cm$^2$) and mean elliptical area of 129 cm$^2$ (24 to 465 cm$^2$). Mean LOD-ratio was 21% (range 3% to 46%) and mean volume ratio was 30% (range 3% to 84%). In 13% of patients, a previously placed mesh had to be removed during the surgery. Surgeries were clean in 87%, and clean-contaminated in 13% of patients. The ACS technique was performed in 35% of all patients and accounted for the reconstructions performed between April 2007 and

**Table 1. Pre-operative conditions seen in the universe of patients undergoing abdominal wall reconstruction for large subcostal hernias.**

| | ACS (n = 11) | TAR (n = 20) | P value |
|---|---|---|---|
| Age (years) | 52 | 58 | 0.5 |
| BMI (kg/m$^2$) | 27.8 | 28.8 | 0.05 |
| Smoking | 27% | 40% | 0.4 |
| Diabetes | 27% | 5% | 0.1 |
| Immunosuppresants | 0% | 5% | 0.7 |
| COPD | 9% | 0% | 0.2 |
| Previous infection or healing by secondary intention | 27% | 15% | 0.4 |
| Previous hernia repair at the same subcostal site | 18% | 35% | 0.3 |
| Hernia vertical size (cm) | 12 | 12 | 0.7 |
| Hernia horizontal size (cm) | 12 | 11 | 0.5 |
| Rectangular area (cm$^2$) | 156 | 166 | 0.5 |
| Elliptical area (cm$^2$) | 123 | 131 | 0.5 |
| V-ratio | 19% | 22% | 0.3 |
| LOD-ratio | 25% | 32% | 0.1 |
| Contamination | 27% | 5% | 0.1 |
| Mesh removal | 18% | 10% | 0.5 |

February 2015. The TAR technique was performed in 65% of patients and this occurred between February 2015 and October 2022. Non-absorbable synthetic mesh was used in all the patients in this series. Median in-hospital stay was 6 days (range 3 to 46 days).

No complications were developed in 61% of patients, while a morbidity rate of 39% was observed. This consisted of local abdominal wall complications in 29% (4 wound infections, 2 seromas, 1 hematoma, 1 wound edge necrosis and 1 sinus tract); intra-abdominal complications in 3% (1 bowel leak); and systemic complications in 19% (2 pneumoniae, 1 acute kidney failure, 1 pulmonary embolism, 1 deep vein thrombosis, and 1 liver insufficiency). After performing statistical analyses, more postoperative local abdominal wall complications were seen in patients with history of previous infection and/or healing by secondary intention in the herniated site (63% vs 17%, p = 0.02), and in patients undergoing ACS as opposed to TAR (55% vs 15%, p = 0.02). More systemic complications were found in patients with history of smoking (45% vs 10%, p = 0.03). In-hospital stay was longer in patients who developed post-operative complications (12 vs 5 days, p<0.0001). Median follow-up was 17 months. Hernia recurrence rate from the entire cohort was 23% and it was more common in patients undergoing ACS as opposed to TAR (55% vs 5%, p = 0.008). Comparisons of pre-operative conditions and post-operative outcomes can be seen in Tables 1 and 2, respectively. Pre-operative and post-operative pictures of a representative case are shown in Fig 2.

**Table 2. Post-operative outcomes in patients undergoing abdominal wall reconstruction for large subcostal hernias.**

| | ACS (n = 11) | TAR (n = 20) | P value |
|---|---|---|---|
| Median in-hospital stay (days) | 6 | 6 | NA |
| Morbidity rate | 55% | 20% | 0.05 |
| Local complications | 55% | 15% | 0.02 |
| Systemic complications | 9% | 25% | 0.3 |
| Total follow-up (months) | 16 | 20 | 0.2 |
| Recurrence rate | 55% | 5% | 0.008 |

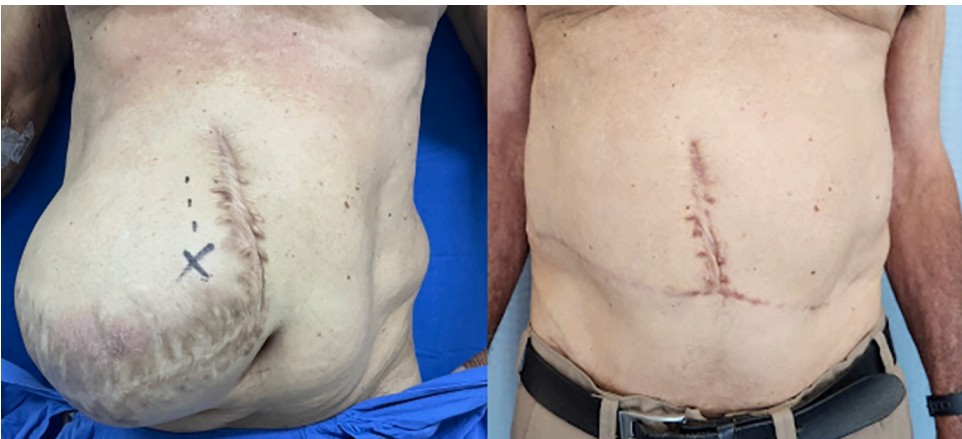

**Fig 2.** (Left) Pre-operative and (right) 1-year post-operative appearance of the same patient.

## Discussion

Subcostal incisional hernias are less common than midline and lumbar incisional hernias [9]. The first scientific report of a subcostal hernia treated surgically dates to 1965, when Argentinian surgeons Zaidman and Alche reported on their novel technique to approach this problem [10]. Since then, a few dozens of other articles have been published in the medical literature describing a myriad of techniques to address this condition. For some time, in this anatomical area meshes were described to be placed either as an onlay, inlay, intra-peritoneally, or in the plane between the external and internal oblique muscles [11–14]. One group reported on the placement of mesh between the transversus abdominis and the internal oblique muscle in patients with subcostal hernias [15]. In these scenarios, mesh fixation is commonly performed either to the abdominal wall muscles using sutures or to the chest wall using either sutures or bone anchors [16,17]. More recently described, the TAR technique allows for the development of a plane between the transversalis fascia and the posterior body of the 12th rib, which can be used for sublay mesh placement with appropriate overlap and without the need for fixation. This plane of mesh placement confers several advantages in the setting of subcostal incisional hernias. First, it is not placed in the subcutaneous space and, therefore, if wound dehiscence occurs, it does not become exposed. Also, it does not elicit the development of chronic seromas. Second, it is not placed intraperitoneally and therefore it does not require to be coated which supposes a more expensive material, and it will not require separation from the bowels in the event of a future laparotomy, a situation that is associated to intra-operative enterotomies [18]. Third, it does not involve dissection of the plane where the motor nerves to the abdominal wall are present. Fourth, the mesh becomes in contact with well vascularized muscle along its entire surface. And fifth, mesh overlap can be achieved adequately. With regards to defect closure, small to medium size subcostal incisional hernias can be usually closed primarily. On the other hand, large subcostal incisional hernias pose a peculiar challenge, since muscle mobilization is impaired in that anatomical site [2]. In the past, we have described the possibility of achieving muscle closure for large size subcostal incisional hernias using the ACS technique, and we used this method between 2007 and February 2015 [6]. Subsequently, we found that the muscle mobilization achieved by the TAR technique can also be applied in the subset of incisional subcostal hernias, and this has become our preferred method since February 2015 [7]. In this study, we compared these two techniques that allow for muscle closure and placement of a large 30x30 synthetic non-absorbable mesh behind the abdominal wall in

patients with large subcostal incisional hernias. With analysis of our data, we have found that in our cohort of patients with large subcostal hernias, the TAR technique performed with a non-coated retromuscular mesh was associated to lower abdominal wall immediate post-operative complications and to lower recurrence rates compared to the ACS technique performed with a coated intra-peritoneal mesh. In terms of lower post-operative morbidity in the abdominal wall, this may be explained by the fact that the TAR technique requires less subcutaneous undermining than the ACS, except in patients with large hernia sacs who develop skin devascularization requiring removal during the surgery. For this reason, we have adopted a direct subcostal approach to the TAR technique which still implies the removal of devascularized skin that results after large hernia sac dissection, but without interfering in the cross-over blood supply from the contralateral side. In terms of the lower recurrence rate seen in patients undergoing large subcostal hernia repair with the TAR technique compared to the ACS, this may be explained by the greater integration that pore meshes experience compared to coated meshes. Also, it is possible that some degree of mesh displacement can be experienced in the long term when meshes are place intra-peritoneally, particularly in the lateral-most edge of the defect.

The muscle mobilization achieved from performing these techniques typically occurs in the lateral to medial direction, which is helpful when trying to close these defects, but it is contrary to the actual axis of muscle separation that occurs when the defects occur after transverse incisions. In this scenario, the muscles separate in a cephalo-caudal direction. Therefore, as proposed by Zorraquino-Gonzalez, a very helpful intra-operative maneuver consists of flexing the surgical table so that the distance between the upper and lower muscle borders is shortened [19]. As in patients undergoing aesthetic abdominoplasty, it is also very helpful to keep a flexed position for some days after the surgery.

The number of patients included in this study precludes the results from being generalized. Future directions in this subset of patients include the use of prophylactic mesh during the index laparotomy both in the elective and in the emergency scenarios [20,21].

## Conclusions

In our cohort of patients with large subcostal hernias, the TAR technique performed with a non-coated retromuscular mesh was associated with lower local post-operative complications and with lower recurrence rates compared to the ACS technique performed with a coated intra-peritoneal mesh.

## Supporting information

**S1 Dataset.**
(XLSX)

## Author Contributions

**Conceptualization:** Antonio Espinosa-de-los-Monteros.

**Data curation:** Daniela Fernandez-Alva, Rodrigo Alejandro Solis-Reyna, Cesar Alberto de-la-Garza-Elizondo, Joseph Vazquez-Guadalupe, Oscar Emmanuel Posadas-Trujillo, Flavio Enrique Diaz-Trueba.

**Formal analysis:** Daniela Fernandez-Alva, Rodrigo Alejandro Solis-Reyna, Cesar Alberto de-la-Garza-Elizondo, Joseph Vazquez-Guadalupe, Oscar Emmanuel Posadas-Trujillo, Flavio Enrique Diaz-Trueba.

**Investigation:** Antonio Espinosa-de-los-Monteros, Oscar Emmanuel Posadas-Trujillo, Flavio Enrique Diaz-Trueba.

**Methodology:** Antonio Espinosa-de-los-Monteros.

**Writing – original draft:** Antonio Espinosa-de-los-Monteros.

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
