## [Decision Letter · Decision Letter 0]

24 Nov 2023

PONE-D-23-19505COMPARISON OF LARGE SUBCOSTAL HERNIAS TREATED WITH OPEN ANTERIOR COMPONENTS SEPARATION OR OPEN TRANSVERSUS ABDOMINIS RELEASE TECHNIQUESPLOS ONE

Dear Dr. Espinosa-de-los-Monteros,

Thank you for submitting your manuscript to PLOS ONE. After careful consideration, we feel that it has merit but does not fully meet PLOS ONE’s publication criteria as it currently stands. Therefore, we invite you to submit a revised version of the manuscript that addresses the points raised during the review process.

We look forward to receiving your revised manuscript.

Kind regards,

Ibrahim Umar Garzali, MBBS, FWACS

Academic Editor

PLOS ONE

Journal Requirements:

Reviewers' comments:

Reviewer's Responses to Questions

**Comments to the Author**

1. Is the manuscript technically sound, and do the data support the conclusions?

Reviewer #1: Yes

Reviewer #2: Yes

2. Has the statistical analysis been performed appropriately and rigorously? 

Reviewer #1: Yes

Reviewer #2: Yes

3. Have the authors made all data underlying the findings in their manuscript fully available?

Reviewer #1: Yes

Reviewer #2: Yes

4. Is the manuscript presented in an intelligible fashion and written in standard English?

Reviewer #1: Yes

Reviewer #2: Yes

5. Review Comments to the Author

Reviewer #1: Tile page

Consider changing the topic to “COMPARISON OF OPEN ANTERIOR COMPONENT AND OPEN TRANSVERSUS ABDOMINUS RELEASE IN REPAIR OF LARGE SUBCOSTAL HERNIAS”

ABSTRACT

Results

Paragraph 3, line 3: “undergoing” changed to “who had”

Paragraph 3, line 4: “undergoing” changed to “who had”

Conclusion

Paragraph 3, line 3: “undergoing” changed to “who had”

MATERIAL AND METHODS

Page 5, line 1: “who underwent” changed to “who had”

CONCLUSION

Page 9 line 2: The word “to” is changed to “with” before the phrase lower abdominal, the phrase “lower abdominal wall” is changed to “local” and the word “to” is changed to “with” before the phrase “lower recurrence”

OVERALL COMMENT

It is a good study with expected result however early postoperative pain and chronic postoperative pain could be considered for future study

Reviewer #2: first of all, the authors need to be applauded for taking the initiative to research this particular study area. the manuscript is well written and the results were well presented. the discussion is apt.

6. PLOS authors have the option to publish the peer review history of their article (what does this mean?). If published, this will include your full peer review and any attached files.

Reviewer #1: **Yes: **Bello Muideen Abodunde

Reviewer #2: No

---

## [Author Response · Author response to Decision Letter 0]

28 Nov 2023

The authors of the paper with new title “COMPARISON OF OPEN ANTERIOR COMPONENT SEPARATION AND OPEN TRANSVERSUS ABDOMINUS RELEASE IN REPAIR OF LARGE SUBCOSTAL HERNIAS” appreciate the reviewers for the observations made. All of them have been addressed and have greatly impacted on improving the overall quality of the document. Reposes are found belos.

1. Please ensure that your manuscript meets PLOS ONE's style requirements. Reviewer #1: Tile page Consider changing the topic to “COMPARISON OF OPEN ANTERIOR COMPONENT AND OPEN TRANSVERSUS ABDOMINUS RELEASE IN REPAIR OF LARGE SUBCOSTAL HERNIAS”

RESPONSE: The title of the study has been changed in the revised version of the manuscript.

ABSTRACT

Results

Paragraph 3, line 3: “undergoing” changed to “who had”

Paragraph 3, line 4: “undergoing” changed to “who had”

Conclusion

Paragraph 3, line 3: “undergoing” changed to “who had”

MATERIAL AND METHODS

Page 5, line 1: “who underwent” changed to “who had”

CONCLUSION

Page 9 line 2: The word “to” is changed to “with” before the phrase lower abdominal, the phrase “lower abdominal wall” is changed to “local” and the word “to” is changed to “with” before the phrase “lower recurrence”

RESPONSE: All these wordings been changed in the revised version of the manuscript.

 RESPONSE: In the revised cover letter we have stated that there are no ethical nor legal restrictions on sharing a de-identified data set. Therefore, we have included a file with the data set that supports the information on the manuscript.

RESPONSE: ORCID ID 0000-0002-6665-5507 has been included in the submission

RESPONSE: The reference list has been reviewed and no further changes have been made.

Antonio Espinosa-de-los-Monteros, M.D.

Chief of Plastic Surgery

Instituto Nacional de Ciencias Medicas y Nutricion 

Mexico City. Mexico

---

## [Editor Report · Decision Letter 1]

14 Dec 2023

COMPARISON OF OPEN ANTERIOR COMPONENT AND OPEN TRANSVERSUS ABDOMINUS RELEASE IN REPAIR OF LARGE SUBCOSTAL HERNIAS

PONE-D-23-19505R1

Dear Dr. Espinosa-de-los-Monteros,

We’re pleased to inform you that your manuscript has been judged scientifically suitable for publication and will be formally accepted for publication once it meets all outstanding technical requirements.

Kind regards,

Ibrahim Umar Garzali, MBBS, FWACS

Academic Editor

PLOS ONE
---

## [Editor Report · Acceptance letter]

18 Dec 2023

PONE-D-23-19505R1 

PLOS ONE

Dear Dr. Espinosa-de-los-Monteros, 

I'm pleased to inform you that your manuscript has been deemed suitable for publication in PLOS ONE. Congratulations! Your manuscript is now being handed over to our production team.

Kind regards, 

on behalf of

Dr. Ibrahim Umar Garzali 

Academic Editor

PLOS ONE